# New n-p Junction Floating Gate to Enhance the Operation Performance of a Semiconductor Memory Device

**DOI:** 10.3390/ma15103640

**Published:** 2022-05-19

**Authors:** Yi-Yueh Chen, Feng-Ming Lee, Yu-Yu Lin, Chih-Hsiung Lee, Wei-Chen Chen, Che-Kai Shu, Su-Jien Lin, Shou-Yi Chang, Chih-Yuan Lu

**Affiliations:** 1Department of Materials Science and Engineering, National Tsing Hua University, Hsinchu 30013, Taiwan; sjlin@mx.nthu.edu.tw (S.-J.L.); changsy@mx.nthu.edu.tw (S.-Y.C.); 2Quality Engineering Center, Macronix International Co., Ltd., Hsinchu 30078, Taiwan; chekaihsu@mxic.com.tw; 3Emergency Central Lab., Macronix International Co., Ltd., Hsinchu 30078, Taiwan; fmlee@mxic.com.tw (F.-M.L.); yylin01@mxic.com.tw (Y.-Y.L.); weichenchen@mxic.com.tw (W.-C.C.); cylu@mxic.com.tw (C.-Y.L.); 4Technology Development Center, Macronix International Co., Ltd., Hsinchu 30078, Taiwan; chlee@mxic.com.tw

**Keywords:** semiconductor device, memory cell, floating gate, n-p junction, charge leakage

## Abstract

To lower the charge leakage of a floating gate device and improve the operation performance of memory devices toward a smaller structure size and a higher component capability, two new types of floating gates composed of pn-type polysilicon or np-type polysilicon were developed in this study. Their microstructure and elemental compositions were investigated, and the sheet resistance, threshold voltages and erasing voltages were measured. The experimental results and charge simulation indicated that, by forming an n-p junction in the floating gate, the sheet resistance was increased, and the charge leakage was reduced because of the formation of a carrier depletion zone at the junction interface serving as an intrinsic potential barrier. Additionally, the threshold voltage and erasing voltage of the np-type floating gate were elevated, suggesting that the performance of the floating gate in the operation of memory devices can be effectively improved without the application of new materials or changes to the physical structure.

## 1. Introduction

Memory devices, one of the most typical and popularly used electronic devices, generally comprise a plurality of gate structures, which include a control gate and a floating gate [1,2]. The floating gate is a conductive layer normally fabricated from polysilicon that is positioned between the control gate and a silicon substrate [1,2]. The floating gate is not attached to any electrodes or power sources and is generally surrounded by an insulation material [1,2]. The operation of the memory cells is dependent upon the charge stored in the floating gate at the threshold voltage to represent information in these memory devices [3,4]. The performance of the memory cells is determined by the programming speed, which is dominated by the speed of the erasing and writing operations [1,2]. The speed is basically limited by the rate at which electrons can be pumped into (writing) and out of (erasing) the devices without causing damage to the device [1,5,6,7]. Typically, writing and erasing operations must be capable of operating within 1 ms at a specified applied voltage [1,6,8,9,10,11].

Aiming at a higher capability but a smaller chip size, the semiconductor industry has been increasingly driven towards smaller and more numerous electronic devices, including memory cells [2,12,13]. To reduce the size and accordingly increase the number of such devices, while simultaneously maintaining or even improving their respective capabilities, the size of components and the distance between such components need to be reduced [2,14,15]. However, as the cell size is reduced, some other issues arise that prevent a further reduction in size [15,16]. One of these issues is that charge leakage from the floating gate may increase, thereby deteriorating the performance of the devices as the individual layers of the gate structures are made smaller and placed closer to each other [15]. In particular, the tunneling oxide will be more seriously damaged with more programming and erasing sequences, resulting in more charge leakage [15]. In order to overcome the issue of charge leakage, many device structures have been proposed, e.g., SONOS, BE-SONOS, TAHOS and 3D FLASH [6,17,18,19,20]. The 3D NAND FLASH structure was proposed as a solution when 2D NAND FLASH reached the scaling limit of a 15 nm process node [21]. Furthermore, the ReRAM [8,22], PCRAM [23,24], FeRAM [25,26] and MRAM [27,28] devices have also attracted much attention in the past two decades as promising candidates for the next generation of nonvolatile memory cells with improved performance. However, new semiconductor devices with markedly shrunken gate structures and reduced charge leakage that do not sacrifice their performance or suffer from environmental contamination are still elusive.

Hence, in this study, two new floating gate structures, including a p-n junction and an n-p junction, were designed, investigated and processed on 300 mm wafers. In these new designs, no extra new material needs to be employed, no new process needs to be developed and no contamination risk needs to be considered when the devices are processed at the semiconductor manufacturing factory. By forming an n-p junction instead of a p-n junction in the first conductive layer (the floating gate), the charge leakage across the second dielectric layer (the inter-polysilicon dielectric layer) may be reduced. This n-p junction interface is anticipated to provide an intrinsic potential barrier to inhibit the leakage path, successfully reducing the charge leakage and enlarging the programming and erasing window. Additionally, upon the reduction of the charge leakage across the second dielectric layer, the second dielectric layer can be made thinner and/or even be completely removed from wrapping the first conductive layer. The gate structure can thereby be made more compact, allowing a smaller semiconductor device without sacrificing the performance of the device.

## 2. Materials and Methods

### 2.1. Device Fabrication

NAND FLASH memory devices with two new floating gate structures were fabricated on p-type 300 mm silicon (Si) wafers with n^+^ junctions. As shown in Figure 1, the memory devices comprise the Si substrate, the first dielectric layer (tunneling oxide, denoted as TUN OX) disposed along the substrate, and the first conductive layer (floating gate, FG) disposed along the first dielectric layer (Figure 1a,c, schematically illustrated from the X- and Y-direction cross-sections, respectively). The second dielectric layer (inter-polysilicon dielectric, IPD) is disposed along the sidewall of the first conductive layer, and the second conductive layer (control gate, CG, such as n-type polysilicon) is afterwards deposited. Two new types of the first conductive layer, i.e., the floating gate, were proposed, including the pn-type (a bottom “p^+^” region followed by a top “n^+^” region) polysilicon and the np-type (bottom “n^+^” followed by top “p^+^”) polysilicon, for which a high-temperature chemical vapor deposition (CVD) boron-doping polysilicon process and a high-temperature furnace phosphorous-doping polysilicon process were applied at 500 °C, in sequence or vice versa. The thickness ratio of the bottom-to-top regions of the pn-type or np-type polysilicon was designed to be around 1:3. For comparison, a conventional floating gate (the control split) was also prepared, with single n^+^ polysilicon as the first conductive layer. The concentration of dopants in the n-type and p-type polysilicon was around 1 × 10^19^ cm^−3^ and 1 × 10^21^ cm^−3^, respectively.

### 2.2. Characterization and Measurement

Thin foils (cross-sectional) of the memory devices around the floating gates were cut by using a focused ion beam system (USA, FIB, FEI Expida1265) and milled with an ultralow current, and the microstructure was observed by using a transmission electron microscope (Netherlands, TEM, FEI Osiris). The depth profile of elemental compositions along the floating gates for understanding the distribution of dopants was determined by using a secondary ion mass spectrometer (France, SIMS, AMETEK ims-6f). The sheet resistance (R_s_) of the floating gates, programing threshold voltage (V_th_) and erasing voltage (V_er_) were measured by using a WAT system (USA, Keysight, 4082F). The charge simulation of the floating gates was performed by the TCAD (Technology Computer-Aided Design).

## 3. Results and Discussion

### 3.1. Microstructure and Chemical Composition

Figure 1b shows the cross-sectional TEM microstructure of the memory cells around the floating gates with np-type polysilicon from the X-directional view. Clearly, the tunneling oxide layer is disposed between the floating gates and the substrate, and the inter-polysilicon dielectric layer is uniformly deposited on the floating gates. The image contrast indicates two regions in the floating gates: the bright region at the top and the dark region at the bottom, and the thickness ratio of the bottom-to-top regions is roughly estimated to be around 1:3. As further illustrated in Figure 1d, the SIMS depth profile along the floating gate confirms four regions of elemental distribution along the floating gate, from top to bottom: (1) the top p^+^ polysilicon for a thickness of about 60 nm, with a silicon element; (2) the bottom n^+^ polysilicon for 20 nm, with silicon and a high concentration of phosphorous dopants; (3) the tunneling oxide and (4) the silicon substrate. It was noted that in region (1), boron dopants were not present due to the improper collection condition of light-ionized boron signals from the uneven film structure of the sample instead of a planar/blanket one. However, the gradually dropping intensity of silicon might reveal the existence of other elements that were very likely boron.

### 3.2. Sheet Resistance and Charge

Figure 2 presents the cumulative probability plot and box plot of sheet resistance for three different floating gates, including the control split and the new pn-type and np-type floating gates. Clearly, the sheet resistance of the new floating gates was higher than that of the control split, (i.e., about 1.7 times for the pn-type floating gates and 2.1 times for the np-type floating gates), which was plausibly caused by the formation of a depletion zone and the narrowed channels for current flow. When a forward bias was applied to the np-type floating gate, or a reverse bias was applied to the pn-type floating gate, a depletion zone of carriers would be formed at the n-p or p-n junction interface [28,29,30,31,32], leading to an open circuit at the bottom region of the floating gates. Current flow was therefore allowed only through the top p^+^ or n^+^ polysilicon paths, respectively, and the narrowed channel would thus result in increased resistance, particularly for the np-type floating gates, as the mobility of holes in the p^+^ polysilicon path was lower than that of electrons in the n^+^ polysilicon path [29,30].

As illustrated in the band diagrams of the neutral and charged states of these three floating gates in Figure 3a,c, different band structures are expected. For the conventional n-type polysilicon floating gate (the control split, Figure 3a) at a programing voltage (positive bias, ΔV) applied to the control gate, the energy band near the control gate will bend downward for ΔV to form a channel near the tunneling oxide for carriers to tunnel through the tunneling oxide into the floating gate for programming [1,2]. The charge in the floating gate depends on the gate coupling ratio (GCR) to influence the efficiency of the device programming [13,33]. In comparison, for the pn-type polysilicon floating gate (Figure 3b) and the np-type polysilicon floating gate (Figure 3c) at a thermal equilibrium state, the Fermi level (E_f_) is close to the valence band in the p^+^ region (conduction by holes) and close to the conduction band in the n^+^ region (conduction by electrons). At a constant Fermi level, the distributions of carriers as well as the energy levels of the conduction band (E_c_) and valence band (E_v_) are thus different in the p^+^ and n^+^ regions at the neutral state, and a depletion zone (a thin region with very few carriers) of high electrical resistance will accordingly be formed at the p-n or n-p junction interface [29,30,31,32].

When a programming voltage ΔV is applied to the control gate, the energy band bends downward, and the carriers will tunnel through the tunneling oxide into either the pn- or the np-type floating gate in the same way as the conventional floating gate. However, owing to the different space charge distributions in the p^+^ and n^+^ regions, electrons stay mainly in the n^+^ region [29,34]. The carriers (electrons) into the pn-type floating gate will induce a reverse bias in the p-n junction to cause the expansion of the depletion zone and the shrinkage of the top n^+^ region for carrier storage, therefore reducing the total stored charge. On the contrary, in the np-type floating gate case, a forward bias will lead to the contraction of the depletion zone and the extension of the bottom n^+^ region for carrier storage, which in turn increases the total stored charge. In addition, because the bottom n^+^ region is close to the tunneling oxide channel and has a low energy barrier for programming, and the top p^+^ region is adjacent to the inter-polysilicon dielectric and has a high energy barrier, the charge leakage of the control gate is expected to be inhibited, which aids in improving the programming efficiency and elevating the programming threshold voltage (V_th_) of the np-type floating gate as investigated below.

Furthermore, the charge simulations given in Figure 3d–f confirm the aforementioned assumption regarding charging in the three different floating gates. When the voltage applied to the control gate (V_g_) is switched from 20 V (the programming state) to 0 V (the retention state), as expected, there is no change in the amount or distribution of charge in the conventional floating gate (the control split, Figure 3d), since the n-type floating gate is simply composed of a single material (n^+^ polysilicon). However, the charge is obviously redistributed, and a part of the charge is lost in the pn-type and np-type floating gates when the gate voltage V_g_ is switched. Clearly, at V_g_ = 20 V, the charge in the n^+^ or p^+^ region of the np-type floating gate is larger than that of the pn-type floating gate. At V_g_ = 0 V, in addition to the fact that more charge in the n^+^ region of the np-type floating gate is retained, a portion of charge in the p^+^ region is retained as well, suggesting that this n-p junction design in the floating gate will benefit the retention of charge, particularly because the p^+^ region is much farther away from the tunneling oxide, making it less likely that a charge leakage will occur.

### 3.3. Threshold Voltage and Erasing Voltage

Two other important factors dominating the programming (writing) window and performance of memory devices include the threshold voltage (V_th_, the gate voltage required to create strong inversion under the gate when the floating gate contains the electrons [35]) and the erasing voltage (V_er_, the voltage required for removing the stored charge (electrons) in the floating gates [36]). When the gate voltage is below the threshold voltage, this device is no longer in strong inversion. This region of device operation is called the “cutoff”, which corresponds to a logical “0” stored in the cell [37]. A higher threshold voltage yields a wider programming window and thereafter benefits more precise control over the read operation state of the devices. For example, two states with programming threshold voltages of 4 V and 2 V define a memory window, ΔV, of 2 V, which is clearly better than a window of 1.5 V attained in the case where the programming threshold voltages of the 0 and 1 states are, respectively, 3 V and 1.5 V. On the other hand, a higher erasing voltage is conducive for a more stable state and more effective retention of the stored charge in the memory devices. However, a higher programming threshold voltage may also cause a more serious impact on the tunneling oxide and induce larger current leakage to lower the erasing voltage.

As mentioned above and presented below in Figure 4, the new types of floating gates, in particular the np-type, are observed to effectively improve the performance of the memory devices fabricated without the application of any new materials or changes to their physical structure. As clearly seen in the cumulative probability plot and box plot, the programming threshold voltage of the np-type floating gate was as high as about 1.2 times that of the conventional one (the control split) and much higher than that of the pn-type one (Figure 4a,b), while the erasing voltage of the np-type floating gate was close to that of the conventional one and also higher than that of the pn-type one, both suggesting the better performance of the np-type floating gate in controlling the operation of the memory devices (Figure 4c,d). The erasing voltages of the floating gates that we showed in Figure 4 were actually measured with a deliberately designed test key to check the floating gate state after the charges of the floating gate were cleaned up by applying a high voltage on the substrate. The lower the |V_er_|, the easier it is for the cell to be turned on, which typically corresponds to a logical “1” stored in the cell. The degradations of the programming and erasing operation (after 3000 cycles) were also investigated to understand the performance of the different types of floating gates, as given in Figure 5. It was clear that the np-type floating gate showed a much better performance than the pn-type one and had the same performance as the control split, indicating no extra current leakage from the tunneling oxide even at a higher threshold voltage.

## 4. Conclusions

In summary, a new np-type floating gate with n-p junction polysilicon (bottom “n^+^” followed by top “p^+^” with a thickness ratio of 1:3) was developed in this study to reduce the charge leakage and improve the operation performance of memory devices. A depletion zone of carriers was formed at the n-p junction interface, leading to a narrowed channel and thus an increased sheet resistance that was 2.1 times that of the conventional floating gate. The relatively high charge storage and retention in the np-type floating gate is able to inhibit the charge leakage, owing to the high energy barrier at the n-p junction interface. Moreover, the programming threshold voltage difference between the 0 and 1 states (i.e., the memory window) of the np-type floating gate was effectively elevated by 1.2 times, while the erasing voltage and its degradation were close to that of the conventional one, indicative of no extra current leakage even at a higher programming threshold voltage and the better operation performance of the memory devices.

## Figures and Tables

**Figure 1 materials-15-03640-f001:**
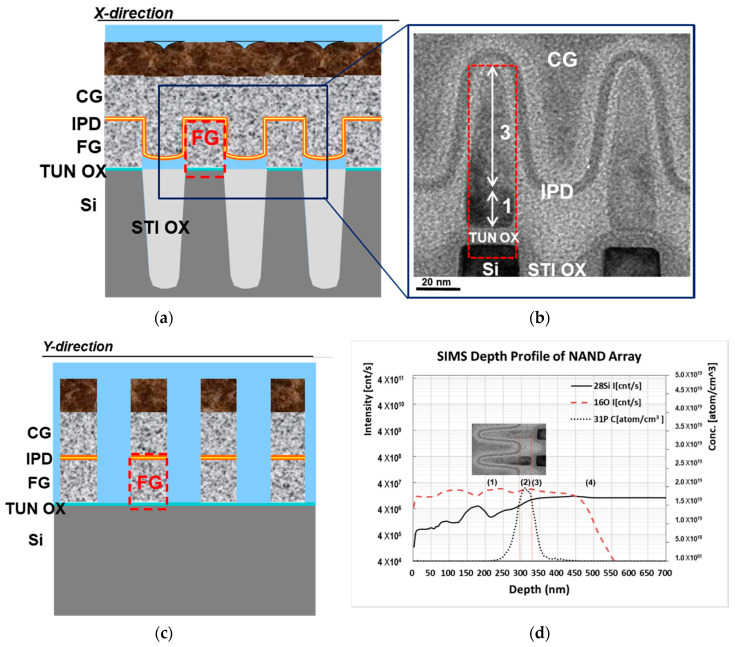
(**a**) Schematic illustration and (**b**) cross-sectional TEM image of memory cells around floating gates with np-type polysilicon from the X-directional view; (**c**) schematic illustration of memory cells from the Y-directional view; (**d**) SIMS depth profile of elemental distribution along the floating gate.

**Figure 2 materials-15-03640-f002:**
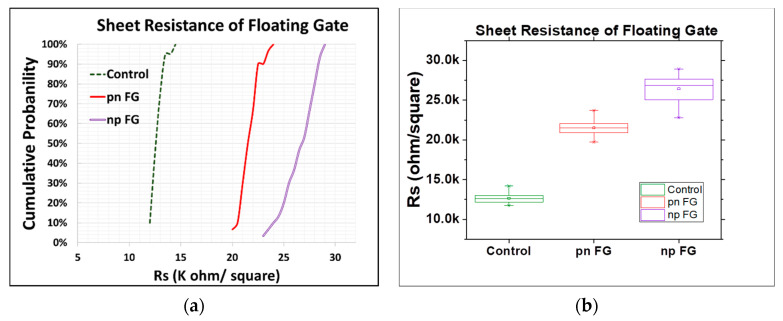
(**a**) Cumulative probability plot of sheet resistance of floating gates (30 data points for pn FG and np FG, and 20 data points for the control split). (**b**) Box plot of sheet resistance (center line: median of the data; top line: Q3, the upper quartile; bottom line: Q1, the lower quartile).

**Figure 3 materials-15-03640-f003:**
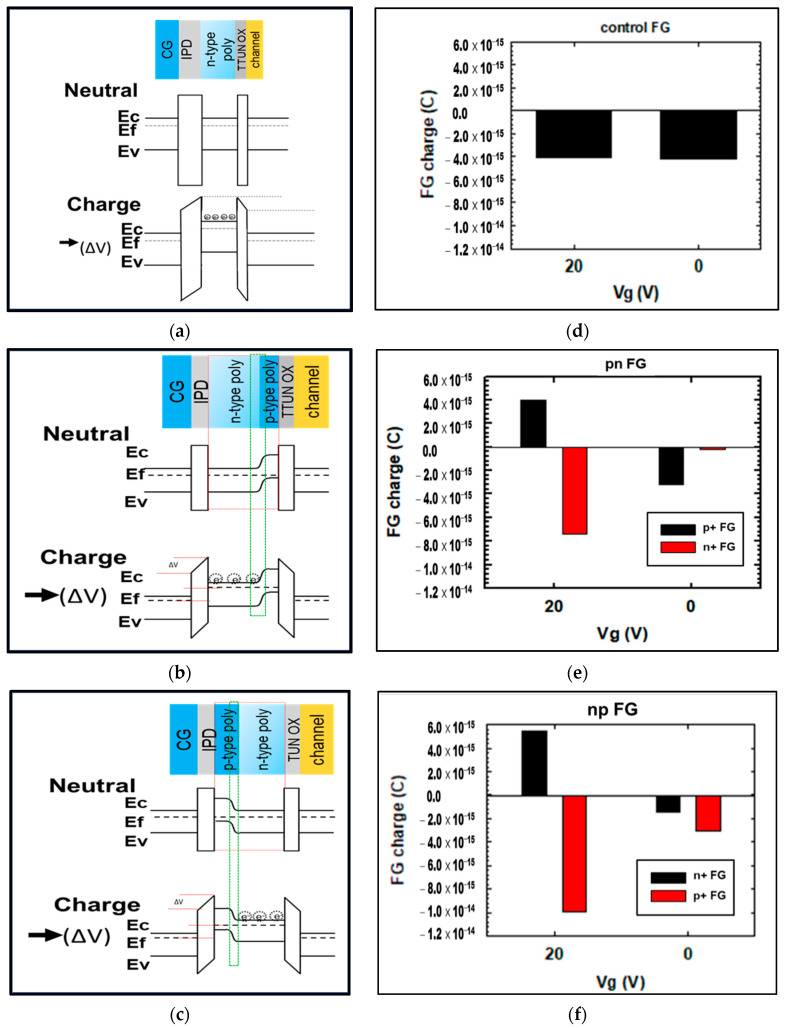
Band diagrams of the natural and charge states of (**a**) conventional n-type polysilicon floating gate (the control split), (**b**) pn-type polysilicon floating gate, and (**c**) np-type polysilicon floating gate. Charge simulations for (**d**) conventional n-type floating gate (the control split), (**e**) pn-type floating gate, and (**f**) np-type floating gate (V_g_ 20 V: programming state, V_g_ 0 V: retention state).

**Figure 4 materials-15-03640-f004:**
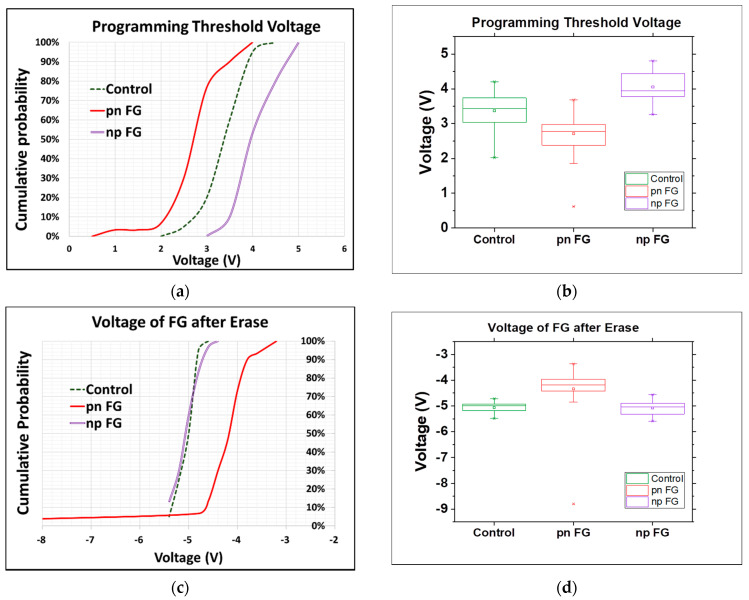
(**a**) Cumulative probability plot of programming threshold voltage of floating gates. (**b**) Box plot of programming threshold voltage. (**c**) Cumulative probability plot of erasing voltage of floating gates. (**d**) Box plot of erasing voltage (center line: median of the data; top line: Q3, the upper quartile; bottom line: Q1, the lower quartile).

**Figure 5 materials-15-03640-f005:**
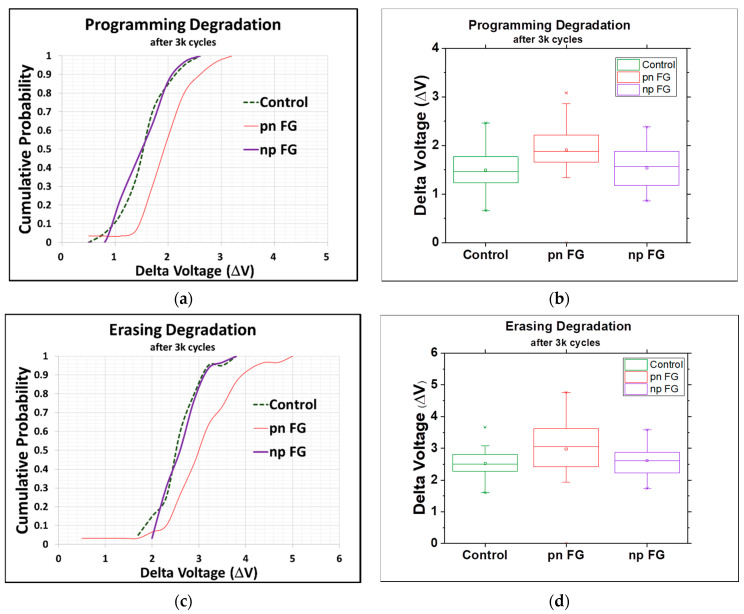
(**a**) Cumulative probability plot of degradation of programming voltage of floating gates. (**b**) Box plot of degradation of programming voltage. (**c**) Cumulative probability plot of degradation of erasing voltage of floating gates. (**d**) Box plot of degradation of erasing voltage (all after 3000 cycles; center line: median of the data; top line: Q3, the upper quartile; bottom line: Q1, the lower quartile).

## Data Availability

The data presented in this study are available on request from corresponding author.

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
