# Peer review of "New n-p Junction Floating Gate to Enhance the Operation Performance of a Semiconductor Memory Device"

_materials, 2022, doi:10.3390/ma15103640_

Round 1
Reviewer 1 Report
Referee report
The article “New n-p Junction Floating Gate to Enhance the Operation Performance of a Semiconductor Memory Device” is devoted to quite actual problem – improving the performance of flash memory. In most commercially used flash memory, the floating gate charge storage material is polysilicon. The authors suggested using n-p junction to enlarge the time of charge storage. The article is interesting and contains new experimental results and can be published. My remarks concern only the editorial correction of the text, so that the text becomes clearer and more accurate.
1) In Figure 1b, it is necessary to set the dimensions along the horizontal and vertical axes (if the scales differ).
2) Figures 2b, 4b, 4d, 5b, 5d – it is not clear, what the dimensions are on the horizontal axis. What are Q1 and Q3 mean?
3) Figures 3a-3f – the numbering of figures in the captions and in the text is incorrect, 3b, 3d, and 3 f are charge simulations, but in the captions and in the text the charge simulations are 3d-3f.
Recommendation: accept
Author Response
Response: We are thankful to the reviewer for the positive and constructive comments. According to the reviewer’s suggestions, the following changes have been made:
1) A scale bar has been added to Figure 1b.
2) The dimensions of the horizontal axis for Figures 2b, 4b, 4d, 5b, and 5d have been updated. The meaning of Q1 and Q3 have been provided in the figure captions as “top line: Q3 the upper quartile, bottom line: Q1 the lower quartile”.
3) The numbering of figures (3a-3f) in the captions has been corrected.
Reviewer 2 Report
Dear authors,
New n-p Junction Floating Gate to Enhance the Operation Performance of a Semiconductor Memory Device is an interesting work which addresses the potential of floating gate n-p junction to perform memory cells. The work is well addressed. The results are clear and readable. The technology used in lately and novel. I recommend at this point to give more information of the technology to the potential reader. As well, the introduction can be improved including other approaches of pn or np junction in the floating gate if available to compare merits. Overall, the work is complete and the conclusions are clear. I suggest the improve of the introduction before being published.
Have the authors evaluated other aspect of the n-p junctions as the hysteresis?
Author Response
Response: We are thankful to the reviewer for the positive and constructive comments. According to the reviewer’s suggestions, the Introduction part has been updated/ improved with the addition of next generation nonvolatile memory research progress, other approaches of new floating gates, and the motivation, originality and merits of the present study on Page 3 to Page 4 as “… In order to overcome the issue of charge leakage, many device structures have been proposed, e.g. SONOS, BE-SONOS, TAHOS, 3D FLASH, etc. [6, 17-20]. Especially the 3D NAND FLASH was proposed as a solution when 2D NAND FLASH reached the scaling limit of 15 nm process node [21]. Furthermore, the ReRAM [8, 22], PCRAM [23, 24], FeRAM [25, 26], and MRAM [27, 28] as promising candidates for the next generation nonvolatile memory with improved performance have also attracted much attention in the past two decades. However, these new semiconductor devices with markedly shrunk gate structures and reduced charge leakage, without sacrificing the performance of the devices or suffering from environmental contamination, are still elusive.” and in the next paragraph “Hence in this study, two new floating gate structures including a p-n junction and an n-p junction were designed, investigated and processed on 300 mm wafers. By this new design, no extra new material needs to be employed, no new process to be developed, and no contamination risk to be consider when the devices are processed at the semiconductor manufacturing factory…”.
As for the hysteresis aspect of the n-p junctions, because the programming Vth of floating gate is affected by the total charge in the floating gate, so the transient behavior of n-p junction will not have hysteresis. In addition, because the read time of NAND is much longer than the relaxation time of NP, it is believed that the hysteresis behavior can be ignored. Although we did no evaluate the hysteresis characteristic of the n-p junctions, we appreciate the reviewer’s suggestion and will consider it in our future study.
Reviewer 3 Report
The present manuscript is certainly significant in terms of the improvement of memory devices. However, suggestions need to be clarified before the publication.
1.The originality of the Introduction part needs to be revised.
2.A few of the references are recent. The authors need to consider citing work that is recent, especially 5 years of recency.
3.Additionally, in Figure 1b, positions for 1, 2, and 3 may be checked, and also spelling mistakes of "degradation" in Figure 5 need to be corrected.
Author Response
Response: We are thankful to the reviewer for the positive and constructive comments. According to the reviewer’s suggestions, the following changes have been made:
- The Introduction part has been updated/improved with the addition of next generation nonvolatile memory research progress, other approaches of new floating gates, and the motivation, originality and merits of the present study on Page 3 to Page 4 as “… In order to overcome the issue of charge leakage, many device structures have been proposed, e.g. SONOS, BE-SONOS, TAHOS, 3D FLASH, etc. [6, 17-20]. Especially the 3D NAND FLASH was proposed as a solution when 2D NAND FLASH reached the scaling limit of 15 nm process node [21]. Furthermore, the ReRAM [8, 22], PCRAM [23, 24], FeRAM [25, 26], and MRAM [27, 28] as promising candidates for the next generation nonvolatile memory with improved performance have also attracted much attention in the past two decades. However, these new semiconductor devices with markedly shrunk gate structures and reduced charge leakage, without sacrificing the performance of the devices or suffering from environmental contamination, are still elusive.” and in the next paragraph “Hence in this study, two new floating gate structures including a p-n junction and an n-p junction were designed, investigated and processed on 300 mm wafers. By this new design, no extra new material needs to be employed, no new process to be developed, and no contamination risk to be consider when the devices are processed at the semiconductor manufacturing factory…”.
- More references (Ref. 17-28) regarding the new development of NAND flash and next generation nonvolatile memory have been added.
3. The numbering (1) to (3) in Figure 5d has been updated as (1) to (4) according to the structure of the memory cell, and their positions have been adjusted ((1) the p+ polysilicon, (2) the n+ polysilicon, (3) the tunneling oxide, and (4) the silicon substrate). Also, the spelling mistakes “degrdation” in Figures 5a and 5d have been corrected as “degradation”.